# Contributions of Common Foods to Resveratrol Intake in the Chinese Diet

**DOI:** 10.3390/foods13081267

**Published:** 2024-04-21

**Authors:** Yichi Xu, Mengxue Fang, Xue Li, Du Wang, Li Yu, Fei Ma, Jun Jiang, Liangxiao Zhang, Peiwu Li

**Affiliations:** 1Key Laboratory of Biology and Genetic Improvement of Oil Crops, Ministry of Agriculture and Rural Affairs; Laboratory of Risk Assessment for Oilseed Products (Wuhan), Ministry of Agriculture and Rural Affairs; Quality Inspection and Test Center for Oilseed Products, Ministry of Agriculture and Rural Affairs; Oil Crops Research Institute, Chinese Academy of Agricultural Sciences, Wuhan 430062, China; 2Hubei Hongshan Laboratory, Wuhan 430070, China; 3College of Food Science and Engineering, Nanjing University of Finance and Economics/Collaborative Innovation Center for Modern Grain Circulation and Safety, Nanjing 210023, China; 4Xianghu Laboratory, Hangzhou 311231, China

**Keywords:** resveratrol, piceid, fruits, dietary intake, vegetables, improvement measures

## Abstract

Resveratrol is a polyphenolic compound with antioxidant and anti-inflammatory properties and therefore has potential health benefits for the prevention and treatment of a wide range of diseases, including cardiovascular disease, cancer, diabetes, and neurodegenerative diseases. The beneficial dose of resveratrol is between 30 and 150 mg. Although the health benefits of resveratrol have been extensively studied, resveratrol intake through the diet of residents in China remains unclear, which restricts the development of resveratrol-rich foods. In this study, a dietary assessment was conducted to reveal that the daily resveratrol intake by Chinese residents through common foods was only 0.783 mg, which was significantly below the beneficial dose. Among the main food types, fruits emerged as the primary source of resveratrol, contributing to 88.35% of the total intake. To improve resveratrol intake, potential methods to increase its consumption were proposed. First method is to increase the resveratrol content of fruits and peanuts. In addition, resveratrol can be extracted from peels. It is also recommended to adopt technical means to improve the bioavailability of resveratrol and develop related supplements and functional drinks.

## 1. Introduction

Resveratrol is a type of phytoalexin produced by plants in response to external influences such as mechanical damage, ultraviolet (UV) exposure, and fungal attack [1]. It was first partially isolated from the bulb of *Veratrum album* in 1939 and later from the root of *Polygonum cuspidatum* in 1963 [2]. *Polygonum cuspidatum* has been used in traditional Chinese and Japanese medicine as an anti-inflammatory and antiplatelet agent. In 1992, Renaud and DeLorgerils first proposed a potential health relationship between wine polyphenols (including resveratrol) and regular or moderate wine consumption, the so-called French paradox [3]. Resveratrol in wine was shown to be a key factor that contributed to the French paradox (high saturated fat intake but low coronary heart disease mortality). This discovery aroused great interest all over the world. Numerous epidemiological and preclinical studies investigated the relationships between resveratrol consumption and human health. They indicated that resveratrol is a promising natural medicine with antioxidant and anti-inflammatory properties, as well as possessing the potential to prevent cancer and cardiovascular disease [4,5].

Resveratrol is the cornerstone molecule of a family that includes glycosides (piceid) and polymers (viniferins). Its fundamental structure features two aromatic rings connected by a methylene bridge, existing in both cis and trans isoforms. These structures have the capacity to attach to glucose, resulting in the formation of cis/trans-piceid. The cis/trans-piceid is a stilbene glycoside compound that releases resveratrol by the action of glycosidase in the gut. It has similar pharmacological effects as resveratrol and can be converted into each other in the body.

Given the health benefits of resveratrol, some people may consider increasing their intake of resveratrol. Grapes and wine are the most common dietary sources of resveratrol, and the content is often related to variety, origin, ripeness, brewing method, and grape fermentation time. In general, the resveratrol content of red wine is higher than that of white wine [6]. Other primary dietary sources of resveratrol include peanuts, mulberries, blueberries, and strawberries [7,8]. In addition, due to the pharmacological effects of resveratrol, supplements and health foods containing resveratrol have been on the market [9]. However, it is unclear whether the body is deficient in resveratrol and whether it is necessary to obtain more resveratrol from the diet and fortified foods. Trans-resveratrol has an established acceptable daily intake (ADI) of 450 mg/day. The ADI is based on a 13-week study that used a no-observed-adverse-effect-level (NOAEL) of 750 mg/kg/day [10]. If more than one gram of trans-resveratrol is taken, it may cause gastrointestinal discomfort, such as diarrhea and nausea, usually within an hour of consumption [11]. Therefore, it is important to evaluate the resveratrol content in individuals’ daily diets.

A dietary assessment is a method to evaluate the nutrients that people consume from different foods in daily diet. It can help us to profile a person’s dietary habits and whether their intake of different nutrients meets recommended intakes. Around the world, there were a number of studies that have used various dietary assessment methods to investigate the dietary patterns and nutrient intakes of different populations. Food frequency questionnaires were used to assess the eating habits of middle-aged and elderly people, and these have shown high reproducibility [12]. Other study used 24 h recalls to keep detailed records of an individual’s diet to assess levels of specific nutrient intakes [13]. In previous studies, we investigated the phytosterol content in edible oils and the effect on phytosterol intake in the Chinese diet, followed by the contribution of tocopherols in common foods to the dietary intake of Chinese residents based on consumption data [14,15]. In addition, the vitamin A content in different foods was used to assess the vitamin A intake in an individual’s daily diet [16].

Currently, only a few studies were conducted on dietary assessments of resveratrol intake. The European Prospective Investigation into Cancer and Nutrition (EPIC) examined data on the composition of resveratrol in common foods in Spain and assessed the primary food sources and their daily intakes in the Spanish adult population [8]. However, systematic studies to assess resveratrol intake in the Chinese population remain rare. In this study, the resveratrol content in different types of food and the resveratrol intake in the daily diet of Chinese people are investigated. In addition, this study provides a theoretical basis for a reasonable intake of resveratrol.

## 2. Materials and Methods

### 2.1. Data Sources

The content and composition data of resveratrol and piceid in coarse cereals beans, vegetables, fruits, and nuts used in this study were extracted from the China Food Composition Table released by the National Institute for Nutrition and Health and Chinese Center for Disease Control and Presentation [17]. This authoritative database compiled representative sample data that encompasses a broad spectrum of varieties and origins, ensuring the reliability and applicability of our research findings. The primary food groups consumed domestically in China were obtained from the Production, Supply, and Distribution (PSD) report issued by the United States Department of Agriculture (USDA), the China Statistical Yearbook, and the China Population Nutrition and Health Monitoring Report [18,19,20]. The consumption of major foods in the Chinese diet, including coarse cereals, potatoes, beans, fruits, and nuts was determined according to the China Population Nutrition and Health Status Monitoring Report and the USDA PSD reports. We selected eight commonly consumed fruits, namely apple, pear, grape, peach, tangerine, orange, grapefruit, and cherry. The fruit consumption data were obtained from the USDA PSD report. In this study, 18 types of foods were evaluated to determine the resveratrol and piceid intake in the Chinese diet.

### 2.2. Calculation Method

The total resveratrol content in this study was calculated using the following equation.
Total resveratrol (μg/100 g) = resveratrol (μg/100 g) + piceid (μg/100 g).

First, the resveratrol and piceid content in different foods were recorded in the Chinese Food Composition Tables, and the total resveratrol content in daily foods was calculated. The total resveratrol content of these foods was then multiplied by the daily consumption of these foods as provided by the United States Department of Agriculture (USDA) and the Chinese Statistical Yearbook to estimate the contribution of the primary types of foods to the total resveratrol intake. Second, to estimate the intake of each fruit required to achieve the beneficial dose of resveratrol, the first step was to determine the beneficial dose of resveratrol and deduct the amount of resveratrol that residents have already consumed through non-fruit sources. The remaining amount of resveratrol required was then divided by the amount of resveratrol in a particular fruit to calculate the amount of additional intake of that fruit. In cases where increasing the intake of only one fruit to meet the beneficial dose of resveratrol should be considered, the resident’s current total intake of resveratrol, including fruits and other food sources, was first subtracted from the beneficial dose of resveratrol. Next, to determine the additional required fruit intake, we divided the required increased amount of resveratrol by the resveratrol content of the target fruit. Finally, this additional intake was added to the current fruit intake to arrive at the new total fruit intake needed to achieve the beneficial resveratrol dose.

Data analysis was performed in Microsoft Excel Version 2021 (Microsoft Corporation, Redmond, WA, USA).

## 3. Results

### 3.1. Resveratrol Contents in Various Foods

Resveratrol is a natural polyphenol compound that is commonly found in plants. Table 1 shows the resveratrol contents in coarse cereals, potatoes, beans, fruits, and nuts. It is clear that fruits contain the highest amount of resveratrol, reaching up to 1941 µg/100 g. For example, tangerine has 1061.43 µg/100 g and peach has 461.6 µg/100 g, demonstrating that tangerine and peach are excellent resveratrol sources. In addition to tangerine and peach, other fruits such as apple, pear, grape, and grapefruit also contain resveratrol, although not as much as tangerine and peach, their resveratrol contents should not be ignored. Apple and pear contain 67 µg/100 g and 34.43 µg/100 g of resveratrol, respectively, while grape and grapefruit contain 79.25 µg/100 g and 82 µg/100 g, respectively. These data suggest that a variety of fruits can serve as a source of resveratrol, helping to enrich the variety of the daily diet. Nuts ranked second in resveratrol content, with walnuts containing 1585 µg/100 g and peanuts containing 74 µg/100 g. Following these, potatoes showed a resveratrol content of 952.4 µg/100 g. Compared with these foods, coarse cereals, and beans had lower levels of resveratrol. The resveratrol content of oats, soybeans, mung beans, and red beans were 56.5, 10.75, 2, and 0 µg/100 g, respectively, highlighting the variation in the resveratrol contents across common food items. These data not only reveal the distribution of resveratrol in different foods but also highlight the importance of getting resveratrol in daily diet through a variety of food choices.

Red wine has been traditionally regarded as a drink rich in resveratrol, and it is a bioactive compound that has been associated with a lower risk of cardiovascular disease in epidemiological studies. The resveratrol concentration in red wine ranges from 1.98 to 7.13 mg/L [21]. According to the beneficial dose, at least 4207 mL of red wine a day would be required to meet the needs of the human body. Obviously, it is not feasible to consume adequate amounts of resveratrol through red wine. Moreover, supplementing resveratrol by drinking superfluous amounts of red wine may lead to excessive alcohol consumption, and this can trigger a series of negative health consequences, including liver disease, cognitive impairment, an increased risk of cancer, and potential alcohol dependence [22,23]. Therefore, while red wine may have some cardiovascular health benefits when consumed in moderation, it should not be relied upon as the primary resveratrol source.

### 3.2. Resveratrol Dietary Intake in the Chinese Diet

Table 2 shows the resveratrol dietary intake among Chinese residents based on the resveratrol contents of five food categories: coarse cereals, potatoes, beans, fruits, and nuts. The total daily resveratrol intake from a resident’s regular diet was 787.74 μg/day. Fruits were the primary source of resveratrol intake, accounting for 88.42% of the total intake. Although fruits did not have the highest resveratrol content (295.97 μg/100 g), they dominated the total intake of resveratrol due to their relatively large consumption (235.33 g/day). Potatoes have a relatively low consumption (5.43 g/day), and they still account for 6.57% of the total intake due to their high resveratrol content (952.4 μg/100 g), ranking second. Nuts had a high resveratrol content at 345.58 μg/100 g. However, due to their lower consumption (10.60 g/day), they ranked third, accounting for only 4.41% of the total intake. Although the consumption of beans was much higher than that of coarse cereals, pulses accounted for only 0.48% of the total intake compared with 0.13% for coarse cereals due to their lower resveratrol content. This highlighted that both the content and consumption of resveratrol are crucial.

Fruits were the primary contributor to resveratrol intake. Table 3 shows the resveratrol content and respective contributions to the overall intake of eight different fruits. The resveratrol intake from fruits was 696.49 μg/day. Tangerine is the most significant contributor, providing 448.12 μg/day of resveratrol, representing 56.89% of the total daily intake. Peach is another significant source of resveratrol, contributing 133.92 μg/day or 17% of the total intake. This highlights the importance of including peach in one’s daily diet and their potential as a resveratrol source. Although apple contains less resveratrol per 100 g than tangerine and peach, due to their higher consumption, they can provide 55.92 μg/day, accounting for 7.1% of the total intake. These data show that even fruits with relatively low resveratrol content can occupy a place in the total intake due to high consumption. Orange, grape, pear, and grapefruit contributed 2.79%, 2.19%, 1.47%, and 0.98%, respectively, to the total intake (21.97, 17.24, 11.58, and 7.74 μg/day). These data revealed that even fruits that are not particularly high in resveratrol still contribute to total resveratrol intake due to their prevalence in daily diets. Cherries had only trace amounts of resveratrol, and their contribution to total intake was negligible, despite being consumed at a rate of 0.71%.

Fruits were the primary contributors to resveratrol intake, accounting for 88.42% of the total. Resveratrol intake may fluctuate significantly at different times of the year due to the seasonal availability of fruit, which can make it difficult for people to consume enough of it, particularly during out season for certain fruits. This fluctuation can affect the health benefits of resveratrol. Although nuts and sweet potatoes can provide a supplementary year-round source of resveratrol, they are relatively low in this compound and cannot fully compensate for the lack of seasonal fruit. The need to develop resveratrol supplements is highlighted by this situation. Resveratrol supplements could offer a reliable and concentrated source of resveratrol, reducing the impact of seasonal variations in fruit availability and ensuring individuals can maintain sufficient resveratrol intake throughout the year.

## 4. Discussion

In recent years, modern medicine and nutrition researches have shown the benefits of resveratrol. It can prevent and treat various chronic diseases. The results indicated a clear correlation between its effect and dose. A daily intake of 30–150 mg of resveratrol is associated with health benefits [24,25]. Our dietary assessment showed that the resveratrol intake of Chinese residents was far from this beneficial dose.

Fruits are a natural source of resveratrol, and they are commonly consumed in the diet. In addition to resveratrol, fruits are also rich in essential vitamins and minerals that promote good health and prevent disease. In this study, it was found that among the five food types in the Chinese diet, fruits contributed the most to the daily resveratrol intake. Tangerines contained the highest resveratrol content, followed by peaches, oranges, grapefruit, grapes, apples, and pears. Table 4 shows the specific amounts of any of the following eight fruits that need to be increased in order to achieve the beneficial dose of resveratrol without changing the intake of other fruits. Tangerine remained the fruit with the highest resveratrol content, with a daily intake of 2794.38 g. To achieve the same resveratrol intake, a person would need to consume up to 84,882.53 g of fruits with lower resveratrol levels, such as pear. According to the Guidelines for a Balanced Diet for Chinese Residents, a daily intake of 200–350 g of fresh fruit is recommended. Meeting resveratrol intake through fruit consumption alone is not feasible due to impracticality for most people and the risk of excessive sugar intake that can increase the risk of diabetes and other metabolic diseases.

Increasing the resveratrol content in fruits is an effective way to increase the intake of this compound. This is also the goal of agricultural producers. Table 5 lists the concentration required for each fruit to meet the beneficial dose of resveratrol when the intake of the other six fruits remains constant. To achieve a beneficial dose of 30 mg of resveratrol, the resveratrol content of tangerines needs to be increased to 70,253.88 μg/100 g, and that of peaches to 101,148.89 μg/100 g. This is 66.19 and 219.13 times the current content, respectively. Therefore, it is crucial to improve breeding techniques to increase the resveratrol content in fruits to meet daily intake recommendations. However, achieving this goal poses a challenge. To increase resveratrol content in fruits, a deep understanding of resveratrol biosynthetic pathways is required. Additionally, genetic variants or bioengineering methods that can effectively activate these pathways must be explored. Furthermore, due to the distinct responses of various fruits to environmental conditions, the breeding process must also guarantee that an elevated resveratrol content does not have any adverse effects on the growth cycle, yield, or other nutrients of the fruit. Moreover, it was found that the resveratrol content in fruits was also influenced by growing environment. It is necessary to classify fruits into different grades according to the resveratrol content. Encouraging consumers to increase their intake of foods containing high amounts of resveratrol but low daily intake is an effective way to increase resveratrol intake. Therefore, it is recommended to gradually increase the intake of these foods with high amounts of resveratrol but currently have a low daily intake, such as mulberries, and incorporate them into the daily diet to increase the total resveratrol intake. Increasing the resveratrol content in fruits through breeding techniques and consuming low-consumption foods rich in resveratrol can promote a diversified and balanced diet, thereby promoting health.

In the research field focused on increasing the resveratrol content in food, peanuts were extensively studied and showed promising results. The resveratrol content in peanuts can be effectively increased through specific post-treatment techniques, such as soaking in water, slicing, exposure to ultraviolet (UV) light, ultrasonic treatment, metal salt treatment, chemical treatment, and exposure to ozone [26,27,28,29]. These methods promote resveratrol biosynthesis by activating defense mechanisms in peanuts. Fungal infections have been found to initiate resveratrol biosynthesis. For instance, when a peanut is sliced and cultured for 48 h in a dark environment at 25 °C, the natural microbiome grows and can accumulate up to 3690 μg/g of resveratrol in the peanut kernels [30]. In addition, research has indicated that peanut germination significantly boosts its resveratrol content. Initially, peanut contains approximately 2.3–4.5 μg/g of resveratrol. However, after nine days of germination, this amount can surge to 11.7–25.7 μg/g, making the resveratrol content in germinated peanuts roughly five to six times higher than in their non-germinated counterparts [31]. Peanuts are a widely grown crop in China and are an important part of the local diet. To achieve a beneficial dose while maintaining the current consumption, peanuts must have a resveratrol content of 3,682.59 μg/g. Therefore, the resveratrol intake can be increased by increasing the resveratrol content of peanuts.

Processing and extracting resveratrol from fruit peels is an effective way to increase the intake of resveratrol. According to the data source of this study, the resveratrol content of red grape skin is as high as 483,420 μg/100 g, which is much higher than the resveratrol content of the flesh at 5020 μg/100 g. However, in their daily diet, people tend to only eat the pulp and ignore the skin and seeds that are rich in resveratrol, which means that a lot of resveratrol resources are wasted. To make more efficient use of these natural resources, measures can be taken to recover these resveratrol-rich pulp peels and extract the substance using scientific methods. Therefore, high resveratrol concentrations can be extracted from grape skins by optimizing the extraction conditions, such as using a specific solvent ratio, temperature, and time [32]. This method can increase resveratrol intake while also efficiently utilizing resources and reducing waste. In addition, this extraction method has potential commercial value. By processing and utilizing agricultural by-products such as fruit peels, not only can resveratrol-rich health products be produced but value can also be added to the agricultural industry chain. For instance, grape skins and seeds can be processed into dietary supplements, functional foods, or cosmetic ingredients. These products are appealing to consumers who seek a healthy lifestyle.

The development of supplements is crucial to increase the intake of resveratrol, as its low solubility in water (approximately 21–30 mg/L) limits its use in foods and beverages [33]. Furthermore, resveratrol undergoes rapid metabolism by the gut microbiome and liver upon oral administration, resulting in the formation of three primary metabolites: resveratrol-3-O-sulfate, resveratrol-4′-O-glucuronide, and resveratrol-3-O-glucuronid. These metabolic processes decrease the amount of resveratrol available for physiological purposes, ultimately reducing its bioavailability [34]. To address these challenges, the researchers utilized various strategies, such as nanoencapsulation technology and other drug delivery systems. Recent research demonstrated that the bioavailability of resveratrol when encapsulated with casein nanoparticles is approximately ten times higher than when administered orally [35]. Other studies have also demonstrated that the oral bioavailability of resveratrol can be significantly enhanced using self-nanoemulsifying drug delivery systems (SNEDDS), solid lipid nanoparticles, and other techniques [36]. Functional drinks containing resveratrol have been launched on the market due to technological advances. Shirai et al. developed a supplement drink that combined resveratrol with other functional ingredients, such as testosterone, L-citrulline, and caffeine, which was shown in clinical studies to improve male sexual function [37]. This finding demonstrates that resveratrol can be effectively integrated into a daily diet to meet specific health needs through scientific formulation and technological innovation. Moving forward, product development can be tailored to the specific needs of different populations, such as a nutritionally fortified milk for children or a sweet juice for adults. This will enable an effective resveratrol intake. Scientific research will continue to drive innovation in resveratrol supplements and functional drinks. Rigorous clinical trials and bioavailability studies can ensure the effectiveness and safety of these products. In summary, using cutting-edge packaging technology and precise formulation strategies, resveratrol intake and bioavailability are expected to be significantly improved. This will not only promote the application of resveratrol in the food and beverage industry but also provide consumers with more healthy choices. This will promote the development of the entire health industry.

## 5. Conclusions

Out of five different food types, fruits contributed the most to resveratrol intake, accounting for 88.42% of the total intake (696.49 μg/day). Tangerine had the highest resveratrol content, accounting for 56.89% (448.12 μg/day) of the total intake. Among 18 common foods, the current daily intake was 0.787 mg, which is significantly lower than the effective dose of resveratrol (30 mg/day). Therefore, it is necessary to increase resveratrol intake under scientific guidance. First, to increase resveratrol intake, individuals can consume fruits that are high in resveratrol but are not commonly consumed. This will not only increase the resveratrol intake but also add variety to the diet. Second, it is crucial to research and develop methods to increase the resveratrol content in fruits and peanuts. Furthermore, extraction of resveratrol from fruit peels can not only enhance the value of the industry chain, but develop supplements to further increase the resveratrol intake. It is also critical to develop resveratrol supplements due to the low water solubility and bioavailability of resveratrol. Nanocapsules and other drug delivery systems could enhance the oral bioavailability of resveratrol. Future product development can be customized to meet the needs of specific populations. Scientific research, clinical trials, and bioavailability studies are crucial to ensure the efficacy and safety of these products. Therefore, this study is of great significance for guiding the Chinese population to supplement their resveratrol intake.

## Figures and Tables

**Table 1 foods-13-01267-t001:** Total resveratrol content of main foods.

Food Categories	Foods	Total Resveratrol Content (µg/100 g)
Coarse cereals	Oat	56.5
Potato	Sweet potato	952.4
Bean	Soybean	10.75
	Mung bean	2
	Red bean	0
Fruits	Apple	67
	Pear	34.43
	Grape	79.25
	Peach	461.6
	Tangerine	1061.43
	Orange	155.33
	Grapefruit	82
	Cherry	0
Nuts	Peanut	74
	Walnut	1585
	Pistachio	0

**Table 2 foods-13-01267-t002:** Contribution of total tocopherol from main kinds of foods.

Food Categories	Consumption (g/Day)	Total Resveratrol Content (μg/100 g)	Intake of Resveratrol (μg/Day)	Percent of Total Intake of Resveratrol (%)
Coarse cereals	1.75	56.5	0.99	0.13
Potatoes	5.43	952.4	51.72	6.57
Beans	39.14	12.67	3.79	0.48
Fruits	235.33	295.97	696.49	88.42
Nuts	10.60	345.58	34.75	4.41
Total			787.74	100.00

**Table 3 foods-13-01267-t003:** Contribution of different kinds of fruits to resveratrol dietary intake.

Fruits	Resveratrol Content (μg/100 g)	Consumption (%)	Intake of Resveratrol (μg/Day)	Percent of Resveratrol (%)
Apple	67.00	35.47	55.92	7.10
Pear	34.43	14.30	11.58	1.47
Grape	79.25	9.24	17.24	2.19
Peach	461.60	12.33	133.92	17.00
Tangerine	1061.43	17.94	448.12	56.89
Orange	155.33	6.01	21.97	2.79
Grapefruit	82.00	4.01	7.74	0.98
Cherry	In trace	0.71	0.00	0.00
Total	696.49			

**Table 4 foods-13-01267-t004:** The amount of fruits for recommended nutrient intake of resveratrol.

Fruits	Resveratrol Content (μg/100 g)	The Amount of Fruits (g)
Apple	67.00	43,683.85
Pear	34.43	84,882.53
Grape	79.25	36,882.65
Peach	461.60	6357.49
Tangerine	1061.43	2794.38
Orange	155.33	18,820.32
Grapefruit	82.00	35,634.15

**Table 5 foods-13-01267-t005:** The increase in resveratrol content in each fruit needed if the daily intake of the other 7 kinds of fruits remains unchanged.

Fruits	Consumption (1000 MT)	Population (1000 Person)	Per Capita Consumption (g/d)	Resveratrol Concentration Required (ug/100 g)	The Current Concentration (ug/100 g)	Multiple
Apple	43,033	141,260	83.46	35,067.61	67.00	523.40
Pear	17,345	141,260	33.64	86,871.05	34.43	2523.23
Grape	11,215	141,260	21.75	134,379.85	79.25	1695.64
Peach	14,959	141,260	29.01	101,148.89	461.60	219.13
Tangerine	21,768	141,260	42.22	70,253.88	1061.43	66.19
Orange	7291	141,260	14.14	206,736.22	155.33	1330.92
Grapefruit	4867	141,260	9.44	309,550.10	82.00	3775.00

## Data Availability

The original contributions presented in the study are included in the article, further inquiries can be directed to the corresponding author.

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
