# Peer review of "Contributions of Common Foods to Resveratrol Intake in the Chinese Diet"

_foods, 2024, doi:10.3390/foods13081267_

Round 1

Reviewer 1 Report

Comments and Suggestions for Authors

The study is interesting and includes an overview of the resveratrol sources used in the Chinese diet. However, the results are based on general data, no individual quantifications were made.

I understand that the main source was the National entity, but at the beginning of the manuscript, the authors declare that the resveratrol content varies depending on multiple factors. This is why I recommend looking at the varieties of the fruits included in the study and trying to find the variety and origin of the fruit, which would be more relevant for further studies.

What type of apples were consumed and if the people tend to peel the apples or not, such facts would also be relevant for the main conclusion.

I do not understand the inclusion of Table 3 in the paper, or at least it is not explained clearly. The caption of this table leads to confusion. Why not suggest reasonable amounts of each fruit to ensure the source varieties and a dietary option adapted for the general population with specific changes (eq. do not peel apples and grapes etc.).

Author Response

The study is interesting and includes an overview of the resveratrol sources used in the Chinese diet. However, the results are based on general data, no individual quantifications were made.

I understand that the main source was the National entity, but at the beginning of the manuscript, the authors declare that the resveratrol content varies depending on multiple factors. This is why I recommend looking at the varieties of the fruits included in the study and trying to find the variety and origin of the fruit, which would be more relevant for further studies.

What type of apples were consumed and if the people tend to peel the apples or not, such facts would also be relevant for the main conclusion.

I do not understand the inclusion of Table 3 in the paper, or at least it is not explained clearly. The caption of this table leads to confusion. Why not suggest reasonable amounts of each fruit to ensure the source varieties and a dietary option adapted for the general population with specific changes (eq. do not peel apples and grapes etc.).

Response: We strongly agree with the suggestion that factors such as the variety and origin of the fruits studied and eating habits (such as whether or not they were peeled) may have an impact on resveratrol content. The sample data used in this study came from the Chinese Food Composition List, which is an authoritative data source. The database collects representative samples covering different origins and various species, ensuring the wide applicability and reliability of the study results. We believe that this is essential to ensure the quality of the research and the legitimacy of the conclusions. Surely, though the data sources are authoritative and representative, the current dataset does not take the effects of processing techniques and eating habits (such as whether or not to peel) on resveratrol content into account. This is a limitation in our study that we will pay attention to and improve in future studies.

Secondly, following the suggestion from the reviewer, we recognize that specific fruit intake in Table 4 might be misleading, because it is impossible to replace other fruits with the one with high content of resveratrol in practice. Therefore, we have removed the original Table 4 (i.e., Table 3 before the changes) and its related descriptions.

Reviewer 2 Report

Comments and Suggestions for Authors

The manuscript entitled "Contributions of Common Foods to Resveratrol Intake in the Chinese Diet" clearly and concisely addresses the importance of consuming foods that contain resveratrol and its health benefits. In the general context, the article is very clear and discusses the percentage of revesratrol found in different foods, as well as what we should consume to obtain adequate percentages of resveratol in our diet. I consider the subject interesting and important for society. The only negative point is that the work focuses only on the Chinese diet. I believe that a more general approach would be more appropriate for this type of journal.

Comments on the Quality of English Language

The manuscript is written in easy-to-understand English.

Author Response

The manuscript entitled "Contributions of Common Foods to Resveratrol Intake in the Chinese Diet" clearly and concisely addresses the importance of consuming foods that contain resveratrol and its health benefits. In the general context, the article is very clear and discusses the percentage of revesratrol found in different foods, as well as what we should consume to obtain adequate percentages of resveratol in our diet. I consider the subject interesting and important for society. The only negative point is that the work focuses only on the Chinese diet. I believe that a more general approach would be more appropriate for this type of journal.

Response: though this work foucses only on the Chinese diet, it provides an approach to assess the resveratrol intake in other conutry. Since resveratrol intake was calculated by multiplying average consumption by average content of each food, it is unmeaningful to resveratrol intake for individual all over the world with different dietary structure and nutriment intake. Conversely, resveratrol intake in local diet should be assessed in future.

Reviewer 3 Report

Comments and Suggestions for Authors

In the submitted manuscript (foods-2934220), the authors analyzed the sources (18 types of food, 8 commonly used fruits) and the amount of resveratrol, a famous antioxidant and anti-inflammatory agent, in the Chinese diet. Considering the obtained (somewhat surprising) results that the average intake of resveratrol is significantly lower than the recommended beneficial doses of this polyphenol, special attention is dedicated to ways to increase the amount of resveratrol in fruit (as its richest source) or to make functional food enriched with this bioactive substance. The findings and recommendations that were obtained represent a valuable contribution to the field of food chemistry.

Kudos to the authors. The work is very well conceived and written. The findings are adequately discussed, with an appropriate comparison with and presentation of the relevant literature. I don't have any substantive objections, only subtle suggestions to bring the text to a "high gloss."

1. Please harmonize the writing of the tested compound's name (everywhere in the text). Therefore, except in the study title, write resveratrol with a small initial letter (e.g., line 211).

2. Check in the instructions for authors whether the number for the literary citation is given before the punctuation mark, for example, before the correct one at the end of the sentence. Usually (from my experience with MDPI journals), it is not.

3. Data in Table 2. The column for resveratrol content in cherries says Tr. I assume it means "in traces." If so, it is better to write it that way.

4. Correct the error (it needs a capital O) regarding the chemical name of the resveratrol metabolite with glucuronic acid (line 22). Next, check the completeness of cited references 9 and 31; the number of pages is missing (lines 343/401).

5. The (substantial) table (marked as S1) was not necessary to be included in the paper supplement since it is certainly not too long. But it's okay this way.

5. Practically, the severe only remark is that nowhere in the text (Materials and methods section) is there a source (even a web address) of how to get data on the composition of food (from the USA and China authorities) that the authors used for their study. Please supplement, i.e., add to the existing text.

Author Response

Reviewer #3 comments:

In the submitted manuscript (foods-2934220), the authors analyzed the sources (18 types of food, 8 commonly used fruits) and the amount of resveratrol, a famous antioxidant and anti-inflammatory agent, in the Chinese diet. Considering the obtained (somewhat surprising) results that the average intake of resveratrol is significantly lower than the recommended beneficial doses of this polyphenol, special attention is dedicated to ways to increase the amount of resveratrol in fruit (as its richest source) or to make functional food enriched with this bioactive substance. The findings and recommendations that were obtained represent a valuable contribution to the field of food chemistry.

Kudos to the authors. The work is very well conceived and written. The findings are adequately discussed, with an appropriate comparison with and presentation of the relevant literature. I don't have any substantive objections, only subtle suggestions to bring the text to a "high gloss."

1.Please harmonize the writing of the tested compound's name (everywhere in the text). Therefore, except in the study title, write resveratrol with a small initial letter (e.g., line 211).

Reply: Following the suggestion from the reviewer, we checked the compound's name in each place and changed it to lower case.

  1. Check in the instructions for authors whether the number for the literary citation is given before the punctuation mark, for example, before the correct one at the end of the sentence. Usually (from my experience with MDPI journals), it is not.

Reply: Following the suggestion from the reviewer, we changed "treat cancer and cardiovascular disease.4, 5." to "treat cancer and cardiovascular disease4, 5."

  1. Data in Table 2. The column for resveratrol content in cherries says Tr. I assume it means "in traces." If so, it is better to write it that way.

Reply: Following the suggestion from the reviewer, we changed "Tr" to "In trace."

  1. Correct the error (it needs a capital O) regarding the chemical name of the resveratrol metabolite with glucuronic acid (line 22). Next, check the completeness of cited references 9 and 31; the number of pages is missing (lines 343/401).

Reply: Following the suggestion from the reviewer, we examined the above issues, and change "resveratrol-3-o-sulfate, resveratrol-4'-O-glucuronide, and resveratrol-3-o-glucuronid." to "resveratrol-3-O-sulfate, resveratrol-4’-O-glucuronide, and resveratrol-3-O-glucuronid.” At the same time we added the page number of references.

  1. The (substantial) table (marked as S1) was not necessary to be included in the paper supplement since it is certainly not too long. But it's okay this way.

Reply: Following the suggestion from the reviewer, we added Table S1 to the article, named it Table 1.

  1. Practically, the severe only remark is that nowhere in the text (Materials and methods section) is there a source (even a web address) of how to get data on the composition of food (from the USA and China authorities) that the authors used for their study.  Please supplement, i.e., add to the existing text.

Reply: Following the suggestion from the reviewer, we added the relevant references.

Round 2

Reviewer 1 Report

Comments and Suggestions for Authors

Table 1 - nuts needs the same formatting as the other groups

Author Response

Table 1 - nuts needs the same formatting as the other groups

Response: following the suggestion from the reviewer, we have revised it.